

# High interspecific variability indicates pollen ice nucleators are incidental

Nina L. H. Kinney[1,2], Charles A. Hepburn[3], Matthew I. Gibson[1,4,5,6], Daniel Ballesteros[2,7], Thomas F. Whale[1,8]

[1]Department of Chemistry, University of Warwick, Coventry, CV4 7AL, UK
[2]Royal Botanic Gardens Kew, Ardingly, West Sussex, RH17 6TN, UK
[3]Mathematics Institute, University of Warwick, Coventry, CV4 7AL, UK
[4]Division of Biomedical Sciences, Warwick Medical School, University of Warwick, Coventry, CV4 7AL, UK
[5]Department of Chemistry, University of Manchester, Manchester, M13 9PL, UK
[6]Manchester Institute of Biotechnology, University of Manchester, Manchester, M1 7DN, UK
[7]Department of Botany and Geology, University of Valencia, 46100 Burjassot, Valencia, Spain
[8]School of Earth and Environment, University of Leeds, Leeds, LS2 9JT, UK

*Correspondence to*: Nina L. H. Kinney (nina.kinney@warwick.ac.uk)

**Abstract.** Ice nucleating molecules (INMs) produced by plant pollen can nucleate ice at warm temperatures and may play an important role in weather and climate relevant cloud glaciation. INMs have also proved useful for mammalian cell and tissue model cryopreservation. The high ice nucleation (IN) activity of some INMs indicates an underlying biological function, either freezing tolerance or bioprecipitation mediated dispersal. Here, using the largest study of pollen ice nucleation to date, we show that phylogenetic proximity, spermatophyte subdivision, primary growth biome, pollination season, primary pollination method, desiccation tolerance and native growth elevation do not account for the IN activity of INMs released from different plant species' pollen. The results suggest that a polysaccharide present in pollen is produced by plants for a purpose unrelated to ice nucleation has an incidental ability to nucleate ice. This ability may have been adapted by some species for specific biological purposes, producing exceptional ice nucleators. Pollen INMs may be more active, widespread in nature, and diverse than previously thought.

## 1 Introduction

Below 0 °C ice is the thermodynamically stable form of water, yet liquid water is routinely observed to persist to temperatures as low as -40 °C because the nucleation of ice is severely kinetically hindered. In nature ice forms usually heterogeneously, stimulated by contact with a surface which can reduce the free energy barrier to nucleation. Of the many substances known to nucleate ice the particles found to induce ice formation at the warmest temperatures are of biological origin (Szyrmer and Zawadzki, 1997; O'Sullivan et al., 2015; Knopf and Alpert, 2023; Cornwell et al., 2023). Ice nucleators associated with representatives from all five kingdoms: animals (Duman, 1982; Layne and Lee, 1995), plants (Gross et al., 1988; Brush et al., 1994; Seifried et al., 2023), fungi (Kieft, 1988; Eufemio et al., 2023), protists (Ickes et al., 2020) and monera (Lindow et al., 1982; Maki et al., 1974; Lukas et al., 2022) have been identified. Expanding knowledge of these ice



nucleators is of biological and atmospheric interest. Better understanding the properties of airborne ice nucleating particles (INPs), and particularly those active at warmer temperatures where secondary ice production mechanisms ensue (Hallett and

Mossop, 1974), is essential to evaluate their atmospheric impact and reduce uncertainties in understanding weather and climate (DeMott and Prenni, 2010; Murray et al., 2012; Kanji et al., 2017; Hummel et al., 2018; Huang et al., 2021; Murray et al., 2021).

Pollen is the male gametophyte of seed-bearing plants. It is established that the pollen of some plants releases water-soluble

ice nucleating molecules (INMs) that can promote ice nucleation (Pummer et al., 2012; Dreischmeier et al., 2017). Pollen INMs readily separate from their parent pollen grains on suspension in water meaning that aqueous solutions of pollen INMs can be prepared straightforwardly (Pummer et al., 2012; Dreischmeier et al., 2017). Both the chemical nature and biological function of these INMs remains unclear, although it seems very likely that they are polysaccharides (Dreischmeier et al., 2017). Improved understanding of the biological function of pollen INMs would facilitate prediction of their likely spatial

and temporal distribution in the atmosphere. There are two obvious hypotheses for the biological function of pollen INMs.

Firstly, intrinsically ice nucleating pollen grains may be more able to survive freezing temperatures, either during development or on dispersal. This would improve their ability to survive freezing conditions to the point of fulfilling their purpose of fertilizing female plant gametophytes. Some plants are known to survive cold temperatures through so-called

'freeze tolerance' where extracellular ice formation at warm temperatures is encouraged (Zachariassen and Kristiansen, 2000). Extracellular ice formation enables intracellular dehydration which favours solidification of cells in a glassy state, facilitating post-thaw survival rather than damaging intracellular ice formation. A similar mechanism may be employed by pollen grains, particularly when they survive drying and the formation of the glassy state (e.g., desiccation tolerant or 'orthodox' pollen). The idea that there may be some relationship between the ice nucleation (IN) activity of pollen and cold

tolerance was first posited by Diehl et al. (2002) and von Blohn et al. (2005). This was based on their observation that high IN activity, of eight pollen samples tested across both studies, seemed to correlate with earlier plant pollination times and therefore pollen transport occurring when it was colder. Pummer et al. similarly noted an apparent dependence of IN activity on geographical distribution and pollination times of plants, with some exceptions (Pummer et al., 2011, 2012). It is worth noting here that pollen INMs have proven to be useful for cryopreservation of mammalian cell suspensions, monolayers and

spheroids, where control of ice nucleation yields greatly improved outcomes (Murray et al., 2022; Gao et al., 2023; Murray et al., 2023). While there are important differences in the biology of animal and plant cells, the fundamental point that avoidance of intracellular ice formation is critical carries across both kingdoms.

Secondly, it may be that pollen INMs favour bioprecipitation, where the INPs generated by pollen promote the formation of

hydrometeors and the distribution of pollen via the water cycle. The concept of bioprecipitation as an ecological advantage has been discussed for other organisms associated with potent ice nucleators (Morris et al., 2004; Kunert et al., 2019). It is



not clear what role these INM play in preserving the viability of pollen grains, which are susceptible to rupture under atmospheric conditions (Grote et al., 2003; Taylor et al., 2004; Wozniak et al., 2018; Subba et al., 2023). Yet, various pine pollens have been shown to retain capacity for germination following immersion freezing (Williams, 2013).


Finally, it should be recognised that pollen INMs may have arisen incidentally, serving some other biological function and nucleating ice by happenstance. The concept of adaptive biological ice nucleators, where IN activity affords an advantage that indicates selectivity in evolution, was explored by Lundheim (2002). Ice-active molecules found in plant tissues control nucleation temperature and ice growth, underpinning mechanisms of frost tolerance or avoidance (Ambroise et al., 2020;

Zachariassen and Kristiansen, 2000). There are examples in nature of ice nucleators present where control over the formation of ice clearly helps to protect against frost damage, such as the presence of ice nucleators in the extracellular space that increase with cold exposure (Zachariassen and Hammel, 1976). While it is difficult to dismiss the possibility that IN activity is an incidental property, it is well known that some biological ice nucleators provide a distinct biological advantage; for others the case is less clear.


To date, pollen samples from only 30 plant species have been tested in immersion mode ice nucleation experiments across the literature, scratching the surface of spermatophyte diversity (Diehl et al., 2002; von Blohn et al., 2005; Pummer et al., 2012; Tong et al., 2015; Dreischmeier et al., 2017; Gute and Abbatt, 2020). Most pollen tested (83%) is considered desiccation tolerant and measurements of desiccation sensitive ('recalcitrant') pollen has been restricted to the grasses

(Poaceae). Notably lacking are measurements of the IN activity of pollen from primarily animal pollinated plant species. Understandably, atmospheric studies have focused on pollen types which are likely to be seasonally abundant in the air and that could impact cloud properties.

## 2. Results and Discussion

We have collected 51 pollen samples from 50 distinct plant species, harnessing the extensive living collection held by the

Royal Botanic Gardens, Kew, UK at their Richmond and Wakehurst sites, and tested their IN activities using a microlitre droplet freezing assay. This instrument, similar to that described by Whale et al. (2015), freezes around 50 one microliter droplets allowing determination of the IN activity of a sample. Previous investigations have focussed mainly on measuring the IN activity of pollen from wind pollinated trees, with some grass species (Gute and Abbatt, 2020; Duan et al., 2023). In this study we explicitly aimed to obtain pollen from a cross section of the plant phylogenetic tree. In doing so, we also

examined species occupying various ecological niches and possessing a range of traits. Table S1 reports details of collections and of the classifications of other plant characteristics reported later in this article.



Droplet fraction frozen curves for five of the tested species are presented in Fig. 1a. Similar plots for all 51 tested samples
are shown in Fig. S1. It can be seen in these two figures that the freezing temperatures for pollen solutions range from as
warm as -5.5 °C to cold temperatures near the homogeneous freezing threshold for one microlitre droplets. These cold
temperature points are in some cases close to the background freezing temperature of the instrument used, shown in Fig. 1a.
Figure 1b. shows the cumulative number of nucleation sites per gram pollen, $n_m(T)$, for all pollen samples, calculated using
Equation (2). Immersion mode ice nucleation measurements of five filtered pollen solutions resulted in mean freezing
temperatures above -10 °C, with the first droplets freezing as warm as -5.5 °C for *Pinus mugo*. Seventeen samples had at
least some droplets freezing above -10 °C (see Fig. S1 and Fig. S2). Only one other sample has previously been found to
have a mean freezing temperature value above -10 °C (Gute and Abbatt, 2020) meaning these observations of high
temperature IN activity are clearly of substantial interest, showing that multiple species can produce highly active INMs. Fig.
1c shows the frequency of occurrence $n_m(T) = 10^5$ g$^{-1}$ across the 51 pollen samples tested. This threshold, henceforth $T_{rep}$,
was chosen as a reasonable response variable for statistical analysis. The choice of this threshold is justified in the methods
section. The majority of species cross $T_{rep}$ in the range of -16 °C to -22 °C with the peak at -19 °C. A considerable number of
species cross $T_{rep}$ at warmer temperatures. To summarise, all pollen solutions tested nucleated ice to some measurable extent.
Some nucleate ice at much warmer temperatures, with many droplets freezing above -10 °C. Pollen from *Musa rubra*, a
species of wild banana, was a cold outlier. We now investigate whether phylogenetic relationships or various environmental
factors can account for the observed variation in ice nucleation temperature between different pollen solutions.

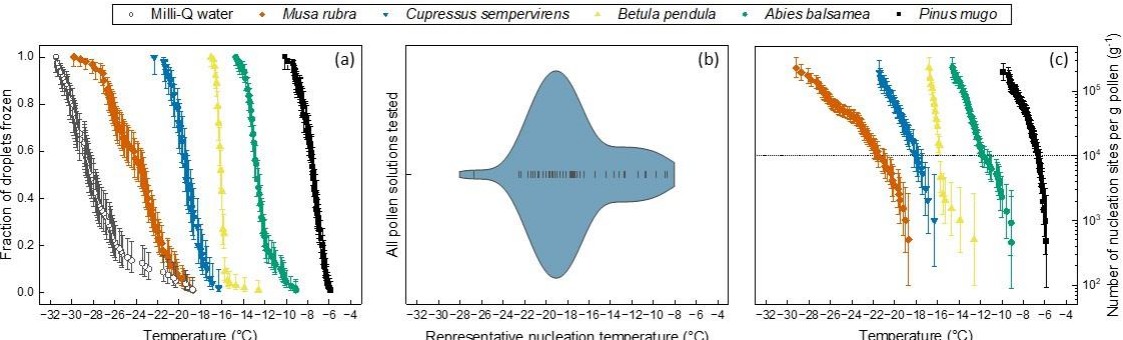

**Figure 1. Summary of the variation observed in the IN activity of pollen samples tested. (a) Fraction of droplets frozen against
temperature for one microlitre droplets of Milli-Q water background and five species' pollen solutions, selected to show the extent
of the variation in activity: *Musa rubra, Cupressus sempervirens, Betula pendula, Abies balsamea* and *Pinus mugo*. (b) Violin plot
showing the distribution of representative nucleation temperature, $T_{rep}$, (T at n$_m$(T) = 10$^5$ g$^{-1}$) of pollen solutions for all 51 samples
tested. (c) Number of nucleation sites per gram pollen, n$_m$(T), against temperature for *Musa rubra, Cupressus sempervirens, Betula
pendula, Abies balsamea* and *Pinus mugo* solutions with dotted line at n$_m$(T) = 10$^5$ g$^{-1}$ corresponding to the representative nucleation
temperature points.**



## 2.1 Phylogenetic study

To establish the phylogenetic relationships between the species tested, we used Plants of the World Online database (Royal Botanic Gardens Kew, 2023) to find accepted species names and phyloT to generate corresponding NCBI tree elements which were then visualised using Interactive Tree Of Life (iTOL). Of the 51 pollen collections in this study, we include results for 50 distinct species in the phylogenetic study. *Cedrus atlantica* f. *glauca* is distinguished from *Cedrus atlantica* at a secondary taxon level, below variety, so is not a distinct species with an individual NCBI number and was therefore omitted from this analysis. For the artificial hybrid *Nymphaea* 'Kew's Stowaway Blues' the NCBI number used for the phylogenetic tree corresponds to a parent of the hybrid: *Nymphaea carpentariae.*

We aimed to measure nucleation temperatures of pollen samples from across the phylogeny. While we have collected samples from representatives of most major plant families, there are some notable absences. Figure 2 shows a phylogenetic tree with the level of IN activity associated with the named species pollen indicated by the length of the blue bars; long blue bar indicates high IN activity and short blue bar indicates low IN activity. The length of the blue bar was calculated from the representative nucleation temperature of the named species' pollen minus the lowest representative nucleation temperature value (*Musa rubra* $T_{rep}$) plus one. This visualisation makes clear that the presence of INMs in pollen is not exclusive to a single plant taxon and confirms it is a more general pollen feature, as suggested by von Blohn et al. (2005). In total, there is no clear relationship between phylogenetic proximity and IN activity of the 50 species' pollen tested in this study.



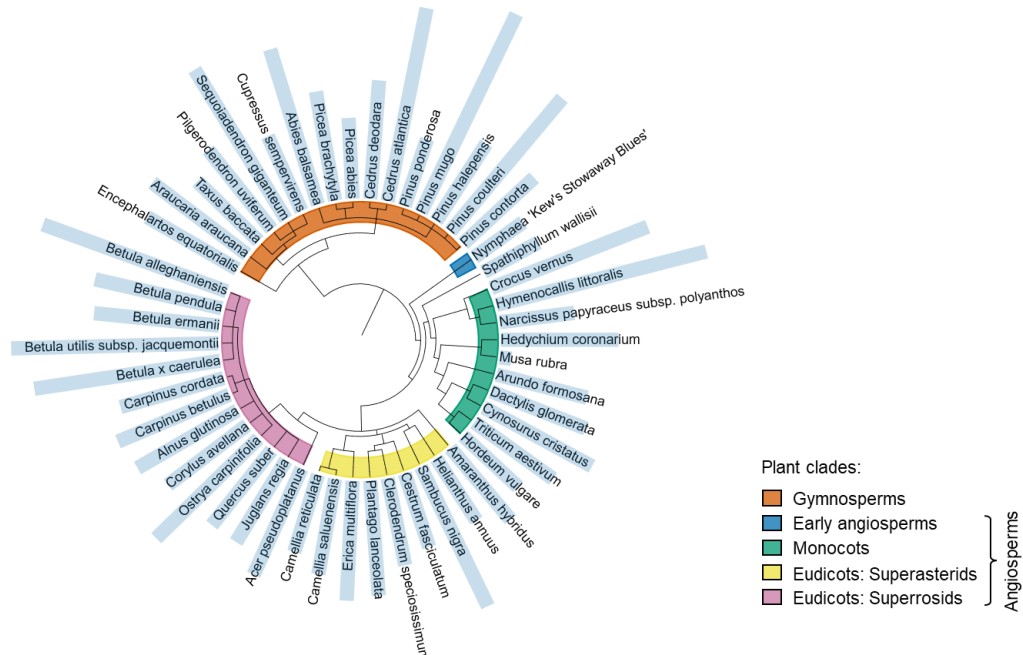

**Figure 2. Phylogenetic tree showing the level of IN activity associated with the named species pollen indicated by the length of the blue bars; short bar indicates low IN activity and vice versa. The length of the blue bar is calculated from the representative nucleation temperature of the named species' pollen minus the lowest representative nucleation temperature value (*Musa rubra* $T_{rep}$) plus one. The main plant clades of species sampled for this study are indicated by coloured arcs. Orange for gymnosperms and other colours for angiosperms: blue (early angiosperms), green (monocots), yellow (eudicots: superasterids), pink (eudicots: superrosids).**


It is at least possible that testing a greater number of pollen samples would reveal trends in activity that the present study does not identify. Given that most of our representative nucleation temperature results fell between -17 °C and -22 °C, as shown in Fig. 1b, it is unsurprising that there is such general consistency in IN activity across the phylogenetic tree. We note however that the more active samples are not clustered in any one area of the phylogenetic tree suggesting that exceptional pollen IN activity is not associated with a specific clade. As closely related plants can occupy quite different ecological niches, it remains possible that environmental factors tend to determine a pollen's ability to nucleate ice. As such, we next divide the set of plants examined by various characteristics to determine whether these impact IN activity of the pollen INMs the plants produce.

## 2.2 Plant characteristics and pollen ice nucleation ability

Figure 3 shows box plots for the 51 pollen samples studied, divided by spermatophyte subdivision, primary growth biome, pollination season, primary pollination method, pollen desiccation tolerance and categorised growth elevation to ascertain



whether these factors influence the IN activity of pollen. The following subsections discuss the dependency of IN activity on individual plant features. In this case our selected plant features, the explanatory variables, have complex relationships and cannot be considered independent. For example, wind pollination is prevalent amongst gymnosperm species and tropical

species are more likely to be pollinated by animals. We therefore did not measure the joint dependency of plant features on nucleation temperature, and instead looked at individual variable influence.

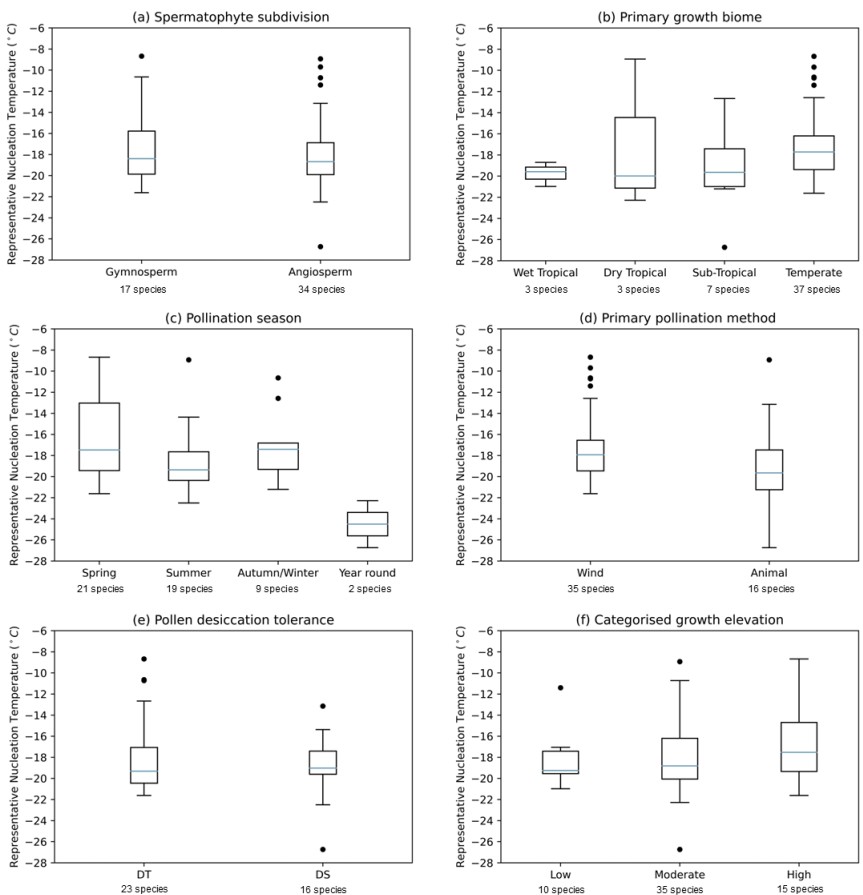

**Figure 3. Boxplots showing the distribution of temperature at $n_m(T)$ = 100000 g⁻¹ for 51 pollen solutions grouped by: (a)**
**spermatophyte subdivision, (b) primary growth biome, (c) pollination season, (d) primary pollination method, (e) desiccation tolerance of pollen grains and (f) categorised growth elevation. The centre line of each box shows the median value for the dataset, the box shows the interquartile range, the whiskers show the maximum and minimum values, while points show outliers.**



### 2.3 Spermatophyte subdivision

The spermatophytes, or seed-bearing plants, are commonly subdivided into two major groups: the angiosperms and gymnosperms (see Fig. 2). Angiosperms, the flowering plants, are the most ecologically diverse group, employing a wide variety of pollination strategies (Bell et al., 2010). Gymnosperms, on the other hand, are non-flowering and are considered to comprise the more 'ancient' lineages (Wang and Ran, 2014). The conifers are the largest of four extant gymnosperm groups, bearing cones which release pollen to be carried by the wind. Figure 3a shows a similar distribution in IN activity between

angiosperm and gymnosperm species. The independent t-test failed to reject the hypothesis that there is a significant difference in mean representative nucleation temperature between angiosperms and gymnosperms (t = 0.673, p = 0.504). This further evidences the variability apparent across the phylogeny of seed-bearing plants. There are properties that are known to have arisen distinctly in separate plant lineages, for example wind-pollination is thought to have evolved at least 65 times (Culley et al., 2002; Stephens et al., 2023), however the apparent ubiquity of these INMs in pollen across all

spermatophytes tested implies that these molecules have been around at least since the divergence of angiosperms and gymnosperms, thought to have occurred between 167 and 199 million year ago (Bell et al., 2010).

### 2.4 Primary growth biome

The biome for each plant species was assigned according to that given in the Plants of the World Online database (Royal Botanic Gardens Kew, 2023). The 50 species tested in this study included representatives from a total of four biomes:

seasonally dry tropical, wet tropical, sub-tropical, and temperate. The artificial hybrid water lily *Nymphaea* 'Kew's Stowaway Blues' was excluded from the boxplot in Fig. 3b. The independent t-test failed to reject the hypothesis that there is a significant difference in mean representative nucleation temperature between sub-tropical and temperate biomes (t = -1.55, p = 0.128), wet tropical and dry tropical groups were omitted from the t-test as these categories contained data for only three species each. The warmest nucleation temperature points appear in the temperate group (see Fig. 3b), however highly IN

active samples from dry tropical and sub-tropical plants were also observed.

### 2.5 Pollination Season

It had been suggested, based on observations in previous studies of an apparent correlation between cold temperature exposure and high IN activity, that a study of various seasonal and geographic pollen sources should be made (von Blohn et al., 2005). We used a host of online sources to assign pollination seasons, corresponding to the pollination times of plants

recorded in their native growth regions, to the 50 species tested. Only two species were classified as pollinating in Autumn, so these were grouped with the Winter species. The ANOVA test failed to reject that there is a significant difference in mean representative nucleation temperature between Spring, Summer and Autumn/Winter pollination seasons (F = 2.10, p = 0.135), the year-round group was omitted from the ANOVA test as this category only contained two species. Again, exceptional nucleators were spread across Spring, Summer and Autumn/Winter groups. Fig. 3c shows no clear difference



between Spring, Summer and Autumn/Winter distributions. Year-round pollinators nucleated ice poorly, however we cannot draw a conclusion from this due to the small number of species in this category.

## 2.6 Primary pollination method

Wind-pollination is prevalent amongst gymnosperms although there are notable exceptions, for instance beetle pollination of *Encephalartos* spp. (Gregory, 1961). Approximately 88% of angiosperm species globally are pollinated by animals, most
usually insects although also by other invertebrates and vertebrates (Ollerton et al., 2011). As such Fig. 3d closely resembles Fig. 3a which divides the dataset by spermatophyte subdivision. The interquartile range representing IN activity is similar for wind and animal pollinated species, however, the lowest nucleation temperature points are found in the animal pollinated group. The independent t-test failed to reject that there is a significant difference in mean representative nucleation temperature between wind and animal pollinated groups (t = 1.61, p = 0.114), therefore we cannot determine a relationship
between IN activity and pollination method. It is not the case that pollens of plants adapted for wind pollination possess greater IN activity, of those we have tested. This strongly suggests that the underlying biological function of pollen nucleating ice is not, or at least not exclusively, a bioprecipitation mechanism.

## 2.7 Elevation range

Figure 3f shows the distribution of representative nucleation temperatures for three plant growth elevation categories. Based
on available data, we categorised plant species primarily by the upper limit of growth elevation recorded for the plants in their native regions. Flexible cut-offs of below 1200 m for 'low', between 1200 m and 2200 m for 'moderate' and above 2200 m for 'high' were used to produce the boxplot. For Fig. 4 we used the numerical values of upper and lower limits of growth elevation for further analysis.

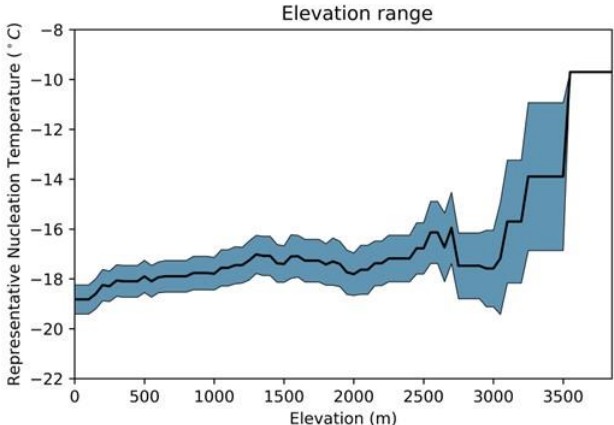

**Figure 4. Plot of possible growth elevation plotted against representative nucleation temperature (T at $n_m(T) = 100000$ g$^{-1}$), the values for which are reported in Table S1. The black line represents the mean across species and the blue shaded area denotes the standard error.**





The ANOVA test failed to reject that there is a significant difference in mean representative nucleation temperature between low, moderate and high elevation categories (F = 0.789, p = 0.460), therefore we cannot determine a relationship between IN activity and these categories. From Fig. 3f we can see a slight increase in the median and variance of representative nucleation temperature and elevation. This relationship is shown in Fig. 4, where representative nucleation temperature is displayed against the possible growth elevation of the plant species. As elevation increases the nucleation temperature slightly increases, until around 3000 m where a sharp increase is observed. Although this indicates a relationship between elevation and nucleation temperature, due to the small number of species sampled which are found growing over 3000 m, more data is needed to substantiate this observation.

## 2.8 Desiccation tolerance

Desiccation tolerance of pollen is a key factor in its longevity and propensity to withstand freezing, so we considered the relationship between desiccation tolerance of pollen grains and IN activity. Categorisation of pollen desiccation tolerance was not possible for all species examined in this study. We categorised the pollen of 39 species as desiccation tolerant (DT) or desiccation sensitive (DS) as outlined in the supplementary information. The assignments for desiccation tolerance of pollen are included in Table S1.

By visual inspection of the distribution of representative nucleation temperatures for 39 species grouped by desiccation tolerance of their pollen, shown in the boxplot Fig. 3e, both DS and DT pollen exhibits a range of IN activities. Pinus spp. pollen is classed as DT, and yet we see a broad range of IN activities across pine pollen samples tested in this study. Results of the independent t-test failed to reject a significant difference in mean representative nucleation temperature between DT and DS pollen (t = 0.996, p = 0.326).

The pollen of plants within the banana family (Musaceae) has been characterised as DS and the pollen of Musa rubra, a low temperature outlier in this study, had a notably high water content on collection. Altogether, this suggests that while no evidence of a direct link between IN activity and desiccation tolerance of pollen grains has been shown here, the water content and biochemistry of the pollen grain, tied to the desiccation sensitivity, clearly has important implications for the INMs released. Therefore, it would be interesting to directly measure the desiccation tolerance and viability of specific pollen samples alongside IN measurements as desiccation sensitivity is a spectrum and is known to vary with stage of development, which may also be an important factor for INM production.

## 2.9 Fourier transform infrared spectroscopy

Fourier-transform infrared (FTIR) spectroscopy was used to gather more information about the soluble components of pollen samples with differing IN activity. Figure 5 shows the normalised FTIR absorption spectra for freeze-dried water-soluble



material from *Pinus ponderosa*, *Betula pendula*, and *Sambucus nigra* pollen. The spectra shown in Fig. 5 are very similar to FTIR spectra results of pollen INMs reported previously (Dreischmeier et al., 2017; Pummer et al., 2013; Felgitsch et al., 2018) and are consistent with polysaccharide absorbances. Overall, the absorption spectra of polysaccharides recovered from different pollen samples are remarkably similar, while small variations are apparent in the fingerprint region, as noted by Gute and Abbatt (2020).

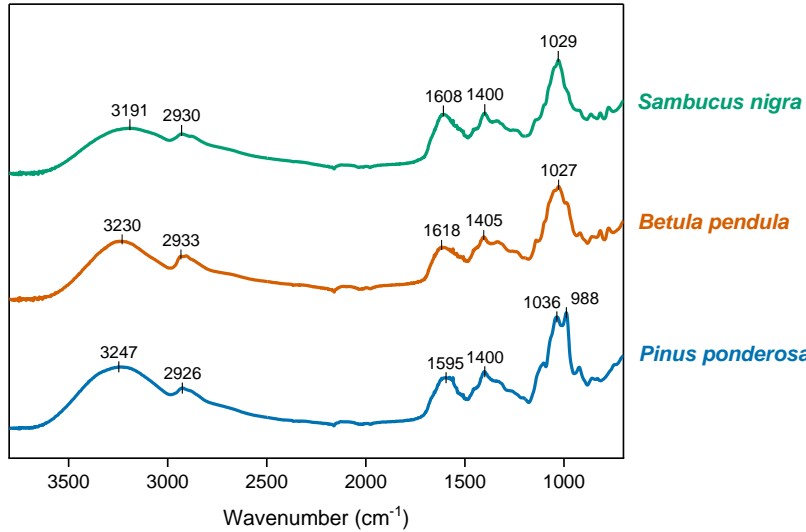


**Figure 5. Fourier-transform infrared absorption spectra for water-soluble material from pollen of three plant species: *Pinus ponderosa* (low IN activity), *Betula pendula* (moderate IN activity), and *Sambucus nigra* (high IN activity). The absorption spectra have been normalised and offset along the y-axis for ease of comparison.**

The *Pinus ponderosa* sample, having the lowest ice nucleation temperature of the three, has a notable difference in the 920-1050 cm$^{-1}$ region when compared with the other two samples. Changes in the 920-1000 cm$^{-1}$ region have been attributed to differences in the configuration of glycosidic linkages (Nikonenko et al., 2005). Gute and Abbatt examined absorption spectra for pollen samples from eleven tree species, comparing with IN activity, and attributed the peak between 965 and 1004 cm$^{-1}$ to the C=C–H bending mode (Gute and Abbatt, 2020). Interestingly, they also observed a weak correlation of this

signal with mean freezing temperature.

The structure of polysaccharides is characteristically complex which renders structural analysis more challenging compared with other biomolecules like proteins (Hong et al., 2021). Diversity in the monosaccharides and linkages comprising polysaccharides may partially explain the extent of the variation in IN activity of different samples. Further work is needed

to better identify the key structures that result in the differences in ice nucleation temperature.



## 2.10 Other observations and caveats

We found, as observed in previous studies, that the IN activity of pollen suspensions (pre-filtration) was largely consistent with the filtered pollen solutions, clearly indicating the separation of the INMs from the pollen grains and insoluble fragments. Results of time series ice nucleation measurements of *Carpinus betulus* solutions, where several pollen

suspensions were filtered over a set of time intervals, indicated that the INMs separate from the grains almost immediately (see Fig. S3). There was no difference in the IN activity of *Carpinus betulus* solutions filtered after <1 hour vs 24 hours in suspension. However, as it is not known where in the pollen grain the INMs originate, and given the huge structural diversity of pollen grains (e.g. ornamentation and number of pores in the exine), it was decided that all pollen samples would be allowed one day in suspension before filtration and ice nucleation measurements, consistent with methods used in previous

studies (O'Sullivan et al., 2015; Pummer et al., 2012; Tong et al., 2015; Augustin et al., 2013; Daily et al., 2022).

We note that ice nucleation experiments are highly susceptible to contamination. The samples we have worked with here are of natural origin and there must be some question regarding contamination from other sources, such as ice nucleating bacteria. However, we note that two collections of *Pinus mugo* pollen were made from plants in distinct locations in different

years and the IN activity of the samples was found to be remarkably similar (see Fig. S4) which suggests such contamination is unlikely, though we cannot completely discount the possibility.

## 3. Materials and Methods

### 3.1 Pollen collection

Most pollen samples were collected from Royal Botanic Gardens, Kew Richmond and Wakehurst sites. Collection methods

were adapted according to plant type and pollen quantity. General methods used were as follows:

For cone bearing plants, the pollen was collected without cutting the plant by covering the male cones with a self-sealing sample bag or glass vial and shaking to release the pollen into the container. For catkin bearing angiosperms, catkins were collected and placed on paper to allow the release of pollen overnight. For other angiosperms, pollen was collected either by

directing the flowers' stamens into a small glass vial and shaking or for flowers with smaller anthers, or lower pollen quantities, stamens were cut and placed on paper and pollen was collected using a small clean spatula to tease the grains free from the anthers. For grass species, the grass culm was cut close to the inflorescence. The cut samples were transported in self-sealing bags and then laid out to dry on paper. Pollen was cleaned by passing it through a 100 μm mesh sieve. Following collection, in all cases, pollen was transferred to a clean glass vial, labelled, and kept refrigerated.




Additional samples to supplement direct collections were purchased from Pharmallerga®. Details of pollen collection are included in Table S1.

### 3.2 Pollen solution preparation

Pollen grains were added to MilliQ® water at a concentration of 2 wt%, or 0.2 wt% where the quantity of pollen collected
would not allow for a sufficient volume of 2 wt% solution for nucleation measurements. Pollen suspensions were refrigerated overnight then filtered using a 0.2 μm pore size filter into a clean glass vial the following day. Ice nucleation measurements of the solutions were conducted following filtration.

### 3.3 Ice nucleation experiments

To measure the ice nucleation activity of pollen solutions arrays of 1 µl droplets were cooled on a purpose-built cold stage,
similar to that described by Whale at al. (Whale et al., 2015), and the nucleation temperature of each droplet recorded. The instrument used for these experiments consists of TEC1-12704 Peltier thermoelectric cooler, connected to a recirculating chiller, driven by a Meerstetter TEC-1091-PT100 Precision Peltier Controller. A $40 \times 40$ mm aluminium plate, on to which a glass slide is placed, is bonded to the Peltier using Arctic Cooling MX-4 thermal compound. Two independent Netshushin PT100 platinum resistance thermometers (NR-141-100S-2-1.0-10-2000PLi-A-3) fixed within the cold stage, read by a
PicoTech PT-104 Platinum Resistance Data Logger, continuously monitor the stage's temperature to within $\pm 0.1$ °C.
Using a Sartorius Picus® electronic micropipette, approximately $50 \times 1$ µl droplets of pollen solution were transferred onto a 22 mm diameter Hampton Research HR3-231 siliconized glass slide, placed on the aluminium plate of the cold stage. The droplets were enclosed by a screw lid top with a window, through which a video camera recorded footage of the droplets as they were cooled. A continuous dry air flow was passed over the droplets to control the relative humidity, preventing frost
growth affecting nucleation temperatures. The cooling (stage temperature and rate of change of temperature) was controlled via a custom LabView program.

For these experiments, the droplet arrays were cooled at a rate of 2 °C/min. The onset of freezing was detected by an increase droplet opacity. The recorded temperature corresponding to the first frame where freezing became visible was taken
as the nucleation temperature of the droplet. Two sets of approximately $50 \times 1$ µl droplet arrays were cooled to freezing for each pollen sample, to enable comparison of IN activity by fraction of droplets frozen against temperature.

### 3.4 Analysis of droplet nucleation temperatures

The fraction of droplets frozen, $f_{\text{ice}}(T)$, is given by:

$$f_{\text{ice}}(T) = \frac{n(T)}{N} \tag{1}$$






where $n(T)$ is the number of frozen droplets and $N$ is the total number of droplets.

The cumulative number of nucleation sites per gram pollen, $n_m(T)$, is calculated from the fraction of droplets frozen, $f_{\text{ice}}(T)$, and the mass of pollen in each droplet of the pollen suspension, $M$, in grams by following equation:


$$n_m(T) = \frac{-\ln\left(1 - f_{\text{ice}}(T)\right)}{M} \qquad (2)$$

So, where the concentration of the pollen suspension is 2 wt% (0.02 g ml$^{-1}$) and the droplet volume is 0.001 ml, the mass of pollen contributing to each droplet $M = 0.02$ mg. $n_m(T)$ can therefore be found, in g$^{-1}$, by:

$$n_m(T) = \frac{-\ln\left(1 - f_{\text{ice}}(T)\right)}{0.0002} \qquad (3)$$


For further analysis, a single temperature value representing the droplet nucleation temperatures of each pollen solution was required. The mean freezing temperature, $T_{50}$, is commonly used for this. In this case, due to very limited quantities of some species' pollen, ten of the 51 solutions tested had to be prepared at the lower concentration of 0.2 wt%. As $T_{50}$ does not account for concentrations, which vary for the solutions tested in our experiments, here we used a representative nucleation

temperature, $T_{\text{rep}}$, which is the temperature value corresponding to $n_m(T) = 100000$ g$^{-1}$. To find $T_{\text{rep}}$ for a given solution we first calculated the number of nucleation sites per gram pollen, $n_m(T)$, values for each recorded droplet freezing event and then calculated the value of $T_{\text{rep}}$ by linear interpolation from the two nearest known points.

To calculate confidence intervals for the data produced we have calculated the Kaplan-Meier estimator for our droplet

freezing experiment using:

$$\hat{S}(T) = \prod_{j|T_j \leq T} \left(\frac{n_u - d_T}{n_u}\right) \qquad (4)$$

where $n_u$ is the number of unfrozen droplets at temperature $T$ and $d_T$ is the number of droplets that freeze at temperature $T$. Note that $f_{\text{ice}}(T)$ is equal to $1 - \hat{S}(T)$. The asymptotic variance of $\ln\left[-\ln\hat{S}(T)\right]$ may be calculated as

$$\hat{\sigma}^2(T) = \frac{\sum \dfrac{d_T}{n_u(n_u - d_T)}}{\left\{\sum \ln\left(\dfrac{n_u - d_T}{n_u}\right)\right\}^2} \qquad (5)$$




where the sums are calculated between $T$ and the first value of $T$ at which a freezing event occurs. The confidence intervals are then given by $\hat{S}(T)^{\exp(\pm z_{\alpha/2}\hat{\sigma}(T))}$ where $z_{\alpha/2}$ is the inverse of the standard normal cumulative distribution, which takes values of 1.96 for the 95% confidence used in this study. Confidence intervals for $n_m(T)$ were calculated by applying Eq. (2) to the absolute values of the upper and lower confidence intervals for $f_{\text{ice}}(T)$. Calculations were performed using Stata 17

(StataCorp, 2021).

### 3.5 Statistical analysis

Representative nucleation temperatures, $T_{\text{rep}}$, (T at $n_m(T)$ = 100000 g$^{-1}$) were used to evaluate the influence of various plant features, our explanatory variables, on IN activity. Independent t-tests and ANOVA tests were used to evaluate whether the groups (e.g. animal pollinated, wind pollinated) within the explanatory variables (e.g. pollination method) showed significant

difference in representative nucleation temperatures. Independent t-tests were used for the variables with two groups and ANOVA tests were used for the variables with more than two groups. These tests were selected based on the distribution of data points and the similarity in variance between groups. The joint dependency of explanatory variables on $T_{\text{rep}}$ was not measured due to the high dependency between explanatory variables. All tests were performed in Python using the scipy.stats package.

### 3.6 Fourier transform infrared spectroscopy

Concentrated (20 wt%) pollen suspensions of *Pinus ponderosa*, *Betula pendula*, and *Sambucus nigra* pollen, purchased from Pharmallerga®, were prepared by adding 1 g pollen to 4 ml Milli-Q® water. After 24 hours in the refrigerator, the suspensions were filtered through 0.2 μm pore size syringe filters. Following filtration, each pollen solution was freeze dried. The infrared absorption spectra of the dried soluble material from the solutions were recorded using the Agilent

Technologies Cary 630 FTIR Spectrometer.

### 4. Conclusions

To the best of our knowledge, this is the most comprehensive survey of ice nucleation by pollen conducted to date. Multiple species, spread across the phylogeny of seed-bearing plants and across ecological niches, can produce INMs that nucleate ice at warm temperatures, above -10 °C. Previously, only one measurement had found pollen INMs nucleating at such warm

temperatures, that of Gute and Abbatt (2020) on *Alnus incana* pollen, however our results suggest that many species' pollen may produce such active ice nucleators.



Almost all pollen types tested produced INMs capable of inducing ice nucleation at moderate temperatures between around -16 °C and -22 °C, as shown by the high-density peak in the centre of Fig. 1c. We found no association between phylogenetic
proximity and IN activity. Exceptional pollen INMs are distributed across lineages. We found some weak trends with environmental factors. Temperate pollens tend to nucleate ice better than tropical ones. Wind dispersed pollens tend to nucleate ice better than animal dispersed ones. Pollen from high elevation plants tends to nucleate ice better than that from low elevation plants. We found none of these trends to be statistically significant, however.

The evidence we have collected suggests that the ice nucleating polysaccharides are near ubiquitous in plant pollen. It seems to us likely that the IN activity of pollen may be conserved incidentally from an ancient ancestor. It is hard to conceive of a reason for polysaccharides produced by the pollen of a tropical, entomophilous plant to nucleate ice. Such a pollen would never be exposed to freezing temperatures. Yet, pollen from *Hymenocallis littoralis* produced notably efficient ice nucleators. Indeed, exceptional ice nucleators are found in pollens from plants occupying all niches investigated.


Classical nucleation theory (CNT) suggests that larger nucleator surfaces are capable of inducing ice nucleation at higher temperatures than smaller ones. Experimentally determined sizes of INMs are found to agree with critical ice cluster size calculated according to CNT (Pummer et al., 2015). This implies that molecules of ~300 kDa must form to account for the observation of ice nucleation above -10 °C by a substantial subset of the pollen solutions we have tested (Pummer et al.,
2015). We suggest that some plants have evolved the ability to produce either aggregates or larger versions of the polysaccharide in order to nucleate ice well for some biological function. It is also possible that larger molecules serve some other purpose and that IN activity is entirely incidental. As some species' pollen which have no conceivable use for effective ice nucleators nevertheless possess them, it seems quite likely that these large polysaccharides or polysaccharide aggregates serve some other biological function, or that such size imposes a negligible evolutionary disadvantage.


We think that the most likely explanation for our results is that a polysaccharide associated with pollen has an intrinsic, incidental, ability to nucleate ice effectively. We hypothesise than some plant species, notably those which live at high elevations, have adapted this polysaccharide to nucleate ice particularly effectively for some biological purpose.

High temperature ice nucleators can have a disproportionate effect on cloud properties as secondary ice production mechanisms are more prevalent at warm temperatures above about -10 °C (Korolev and Leisner, 2020). We propose that pollen should now be considered as a possible source of highly active INPs in the atmosphere. Previous studies have quoted an estimate of global pollen emissions to the atmosphere of 84.5 Tg year[-1], based on a generalisation of pollen grains released by a single species, *Zea mays* (Jacobson and Streets, 2009). Hoose et al. using an aerosol-climate model
incorporating a simplified emission parametrisation based on this estimate found pollen, considered together with other primary biological aerosol particles, has little impact on atmospheric ice nucleation on a global scale (Hoose et al., 2010b, a).

It has been suggested that regional impact may be far greater (Williams and Després, 2017). More recently, studies have considered the potential for sub-pollen particles (SPPs), generated by pollen rupture under atmospheric conditions (Steiner and Solmon, 2018; Subba et al., 2023), to act as persisting carriers for pollen INMs compounding their atmospheric relevance (Burkart et al., 2021). Due to their size, SPPs can accumulate in the atmosphere to a greater extent than whole pollen grains; simulations estimate SPP concentrations are 4-6 orders of magnitude higher than pollen grain concentrations (Werchner et al., 2022). Given that SPPs can significantly influence ice particle number density, impacting mixed phase cloud composition and precipitation, when considered as highly efficient ice nuclei (Werchner et al., 2022), the possibility that a consequential proportion of pollen material holds highly active ice nucleators must now be considered.

**Author contributions**

NLHK, CAH, DB and TFW contributed to pollen sample collection. NLHK conducted ice nucleation experiments and performed data analysis. CAH advised on and conducted statistical analysis. MIG, DB and TFW supervised the project. NLHK and TFW wrote the original manuscript draft. All authors provided critical feedback on the manuscript.

**Competing interests**

TFW, MIG and NLHK are named inventors on a patent application relating to this work.

**Acknowledgments**

NLHK thanks the Natural Environment Research Council and the CENTA Doctoral Training Partnership for a PhD studentship (NE/S007350/1). TFW thanks the Leverhulme Trust and the University of Warwick for supporting an Early Career Fellowship (ECF-2018-127). MIG thanks the European Research Council (ERC) under the European Union's Horizon 2020 research and innovation programme (grant agreement n° 866056) and the Royal Society for an Industry Fellowship (191037) joint with Cytiva. The Royal Botanic Gardens, Kew, receive grant-in-aid from the UK government (Department for Environment, Food and Rural Affairs).

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
