# Peer review of "High interspecific variability in ice nucleation activity suggests pollen ice nucleators are incidental"

_EGUsphere, 2023_

## Author Comment (AC1)

**Responses to RC1**

We thank Professor Morris for her thoughtful comments and suggestions for improving the manuscript. The specific comments made (blue text) are each addressed below (black text) with corresponding changes to the manuscript included (red text).

Title: The title is not clear. Firstly, one might ask what variability. Secondly, it would be more appropriate to say "suggests" rather than "indicates". A better title could be "High interspecific variability in ice nucleation activity suggests pollen ice nucleators are incidental"

Thank you – we agree and intend to change the article title to that suggested.

L 30. This is an outdated taxonomy. There are 3 domains of life: Eukaryotes, Prokaryotes and Archaea. Within the Eurkayrotes there are 4 kingdoms: plants, animals, eumycota (true fungi) and the Stramenopila (formerly Chromista) that contain flagellated organisms. And, there are also the "non living" organisms, i.e. viruses, for which some have wondered about their ice nucleation activity. Taxonomy is always changing (see wikipedia for a summary of this change: https://en.wikipedia.org/wiki/Kingdom_(biology)). The number and names of the taxonomic divisions is not really pertinent to this work. I recommend simply stating that ice nucleation is wide spread across many different types of organisms and indicate examples.

We will replace the relevant section (L 29-33):

"Ice nucleators associated with representatives from all five kingdoms: animals (Duman, 1982; Layne and Lee, 1995), plants (Gross et al., 1988; Brush et al., 1994; Seifried et al., 2023) fungi (Kieft, 1988; Eufemio et al., 2023), protists (Ickes et al., 2020) and monera (Lindow et al., 1982; Maki et al., 1974; Lukas et al., 2022) have been identified. Expanding knowledge of these ice nucleators is of biological and atmospheric interest."

with the following:

"The identification of ice nucleators associated with various organisms, from bacteria (Lindow et al., 1982; Maki et al., 1974; Lukas et al., 2022) to lichens (Kieft, 1988; Eufemio et al., 2023) and plants (Gross et al., 1988; Brush et al., 1994; Felgitsch et al., 2018), is of biological and atmospheric interest."

L 39. This section describes pollen in terms of its chemistry. However, the authors suggest that knowledge about INA of pollen would have implications for atmospheric science. Hence, the authors should also tell the reader about the capacity for pollen to become airborne. Indeed, is all pollen meant to fly? Overall, the authors should introduce information about the biology and "life cycle" of pollen.

The following section will be added to the text (L 39) to better introduce different pollination strategies and the capacity for pollen to become airborne.

"Pollen grains are the male gametophytes of the seed plants or spermatophytes, a clade that is divided into gymnosperms (plants with unenclosed or "naked seeds") and angiosperms (or flowering plants). Angiosperms predominantly rely on mutualistic relationships with animal pollinators for transporting pollen to the female stigma. There, pollen develops a pollinic tube that transports the spermatic nuclei into the ovule, where the female gametophyte is located

and fertilisation occurs, enabling the production of seeds. Anemophily (wind pollination) is the primary pollination method for most gymnosperms and around 10% of angiosperm species (Friedman and Barrett, 2009). Seasonally, anemophilous plants release large quantities of pollen into the air. Pollen grains and smaller pollen fragments are classed as primary biological aerosol particles and are known to disperse in the atmosphere (Després et al., 2012). Records of the long-distance transport of pollen and its presence in rain, hail and snow samples have appeared in historical literature (Bessey, 1883; Potter and Rowley, 1960; Gregory, 1961), pointing to the possible relevance of pollen for ice nucleation in mixed-phase clouds."

L 39. Furthermore, in this section, here the authors indicate that ice nucleation activity is assessed from solutions of material that is washed from the pollen grains. This information needs to be emphasized to help readers understand that ice nucleation activity will NOT be assessed strictly on a per pollen grain basis via tests of suspensions of pollen grains. I had not paid attention to this detail in my first round of reading the manuscript and it threw me off – especially given that the Materials and Methods section comes after the Results section and that I have the unfortunate habit of reading a manuscript from beginning to end in the order that it is presented. Here in the introduction section it would be very useful to present information on the amount of INA material released per grain and if this is constant among plant species. This is critical information because, for pollen INA to be pertinent for the life history of a plant, it needs to function on a whole grain. In addition, even if the material in the washed-off form is active in cloud processes, it is unlikely that its distribution in the atmosphere is homogenous – it is more likely to follow the distribution of the pollen grains themselves. If the amount of INA material per g (or per grain) of pollen is very different among species, how can this be accounted for in the analyses and can the results be expressed on a per grain basis to allow for pertinent comparisons?

We thank Prof. Morris for highlighting that the method for preparing the pollen solutions should be more clearly emphasised earlier in the manuscript. We will clarify this in the Introduction – the paragraph beginning at L 39:

"It is established that the pollen of some plants releases water-soluble ice nucleating macromolecules (INMs) that can promote ice nucleation (Pummer et al., 2012; Dreischmeier et al., 2017). Pollen INMs readily separate from their parent pollen grains on suspension in water meaning that aqueous solutions of pollen INMs can be prepared straightforwardly (Pummer et al., 2012; Dreischmeier et al., 2017). Both the chemical nature and biological function of these INMs remains unclear, although it seems very likely that they are polysaccharides (Dreischmeier et al., 2017). Improved understanding of the biological function of pollen INMs would facilitate prediction of their likely spatial and temporal distribution in the atmosphere. There are two obvious hypotheses for the biological function of pollen INMs."

will be changed to the following:

"It is established that the pollen of some plants release water-soluble ice nucleating macromolecules (INMs) that can promote ice nucleation (Pummer et al., 2012; Dreischmeier et al., 2017). The IN activity of pollen grains in suspension is found to agree with the IN activity of pollen solutions following removal of grains and insoluble fragments by filtration (Pummer et al., 2012; Dreischmeier et al., 2017; Gute and Abbatt, 2020), indicating that pollen INMs readily separate from their parent pollen grains on contact with water. The IN activity of pollen samples can therefore be evaluated by measuring nucleation temperatures of filtered pollen solutions. The quantity of soluble material in a filtered pollen solution has been recorded for a handful of pollen types (Pummer et al., 2012; Murray et al., 2022) but the proportion of this

material that nucleates ice is estimated to be far smaller (Murray et al., 2022). Both the chemical nature and biological function of these INMs remains unclear, although it seems very likely that they are polysaccharides (Dreischmeier et al., 2017). Improved understanding of the biological function of pollen INMs would facilitate prediction of their likely spatial and temporal distribution in the atmosphere. There are two obvious hypotheses for the biological function of pollen INMs."

We will also add the following sentence at the start of the Results and Discussion section (L 92):

"We evaluate the IN activity of pollen solutions (also commonly referred to as pollen washing waters) which are prepared by filtering aqueous pollen suspensions, removing pollen grains and insoluble fragments and leaving only soluble extracts in solution."

The concept of determining per grain activity is discussed further below, in response to the comment regarding L 356. Measurements of soluble material is discussed further in response to the comment on L 387.

L 84. The authors have written "measurements ...... has....". Please correct the grammar.

We will correct this, thanks.

L 85. Here the authors mention pollination by animals. Some readers might be lost if there is not a presentation of the different types of pollen, as suggested above.

In our updated manuscript we will introduce this concept in the Introduction as suggested (see L 39 response).

L 216. Here the authors explain that their results do not support that the "underlying biological function" of pollen nucleating ice is a bioprecipitation mechanism. It might be more precise to state that their results do not support that "there is a positive selection pressure - for pollen in general - of rainfall on ice nucleation capacity. Ice nucleation might indeed be involved in depositing pollen from the atmosphere in some cases, but their results show that there is no signal of positive selection pressure for this across all of the different types of pollen and not for the wind-disseminated pollen in particular.

Agreed. We intend to replace the final sentence of this section (L 216-217):

"This strongly suggests that the underlying biological function of pollen nucleating ice is not, or at least not exclusively, a bioprecipitation mechanism."

with the following:

"While ice nucleation may be involved in some cases in depositing pollen from the atmosphere, these results do not support that there is a positive selection pressure of precipitation on IN activity of pollen in general."

L 291-296. In this paragraph the authors indicate that the microbiology of the pollen could be a factor in the ice nucleation activity. For pollen collected from outdoor sources, this is a real possibility. The authors could mention how they could have accounted for this factor i.e.

microbiological analyses). Furthermore, the comparable results for filtered and raw suspension is a strong suggestion that the INA they measure is not due primarily to the microflora of the pollen.

To better address these points the following section will be added to the text at L 296.

"Pollen solutions prepared from purchased pollen samples have previously been shown to be sterile with no microbial growth detected on agar jelly (Murray et al., 2022). Comparable results for filtered and unfiltered samples, where measured, also supports that the IN activity represented is not primarily due to the microflora of the pollen. However, because the majority of pollen samples used in this study were collected from outdoor sources, we cannot entirely discount the possibility that microbiological contamination may have contributed to the IN activity of some samples. As such, microbial analysis of pollen samples might be of value in the future."

L 333. The authors state that droplet arrays were cooled at a rate of 2 °C/min. Why this rate of cooling? Does it correspond to the rate that the pollen INA materials would encounter in nature?

A constant cooling rate of 2 °C/min is in the range of likely cooling rates that water droplets in mixed-phase clouds might experience but hold no particular physical significance. This cooling rate is standard for microlitre droplet freezing experiments; it is slow enough to allow for accurate determination of droplet nucleation temperatures using this kind of set-up and fast enough that experiments can be conducted within a reasonable timeframe in the lab.

The following sentence will be added to L 333.

"This cooling rate was chosen for convenience and is similar to that used in many previous studies e.g. Whale (2024)."

L 356. The authors calculated the number of nucleation sites per gram pollen. Is it possible to calculate per pollen grain? Is the weight of pollen homogenous across plant species? Furthermore, is the weight of pollen stable for any given plant species or is it variable depending on environmental conditions? I would expect these topics to be addressed somewhere in the manuscript.

It is important to emphasise that in all studies of IN activity of pollen to date the nucleation temperatures of droplets of pollen suspensions (unfiltered) are found to be consistent with the nucleation temperatures of droplets of pollen solutions (filtered), as discussed in Section 2.10.

When pollen grains are added to water the INMs seem to completely separate from their parent grains and disperse in solution (Pummer et al., 2012; Burkart et al., 2021). For this reason, the activity of filtered pollen solutions is representative of the pollen grain activity, in environments where pollen grains encounter water. Filtered pollen solutions are preferable for conducting droplet freezing experiments for ease of handling and in detecting the change in droplet opacity corresponding to nucleation temperature; in cloudy, typically yellow, pollen suspensions with clumps of grains the change is less clearly visible. Filtration also has the advantage of removing larger potential contaminants.

The mass of pollen is not consistent across species; the grain size and shape, exine thickness, water content and biochemical composition will differ between species as well as during pollen development. Generally, the pollen of anemophilous plants is lighter, facilitating wind dispersal,

and that of animal pollinated plants heavier. We cannot therefore calculate the number of nucleation sites per pollen grain directly from the mass of collected pollen. Pollen grain mass is also often not recorded in online pollen databases (such as https://www.paldat.org/).

The quantity of INMs generated by an individual pollen grain will be highly variable, depending exposure to conditions (e.g. humidity) and atmospheric lifetime. In principle an estimate of number of pollen grains in a given mass could be made by counting grains for each pollen type, however it would be difficult to interpret a meaningful 'per grain' activity relating to the potential activity in nature directly from this.

L 387 and onward. In the Conclusion, could the authors discuss the plasticity of plants in terms of the amount of INA polysaccharide that it produces on pollen grains?  Is this is stable trait? Is it susceptible to environmental conditions and which ones?  This could give insight into the differences observed between species even if the role of the trait is incidental concerning ice nucleation per se

The questions highlighted here are particularly pertinent and the answers not yet known but we have included some discussion below and intend to address this in the updated manuscript (Section 2.10), as suggested, to further reflect on these ideas. Ultimately, this presents a great challenge to measure and we cannot currently explain why the quantity of INMs varies per pollen grain or between pollen types. Learning more about the molecules responsible for this activity will help to shed light on this.

We can, given sufficient quantities of pollen, measure the quantity of soluble material in a filtered pollen solution by freeze-drying and then weighing the dried component as highlighted by Prof. Grothe. This allows for the mass of soluble material generated from a given mass of pollen to be determined. Although it does not provide information on the proportion of the soluble material that is the active component, this potentially better represents the quantity of INMs in each solution than the mass of pollen used for the starting suspension does.

The mass of soluble material has previously been measured for *Betula pendula* and *Carpinus betulus* pollen. It was estimated that ~37% of *B. pendula* pollen mass and 33% of the *C. betulus* pollen mass was soluble. In the current study we have not measured the mass of soluble material due to the limited quantities of pollen obtained from our collections. The material that remained after drying solutions of *Pinus ponderosa*, *Betula pendula*, and *Sambucus nigra* which we used for FTIR absorption measurements was too little to weigh on a standard lab balance. A higher volume of starting material would be needed to determine the percentage of pollen mass of the soluble components.

It would be interesting to compare the amount of soluble material generated from more diverse pollen types but the implications are limited in that the relative proportion of INMs within this quantity is unknown. These solutions contain a complex mix of proteins and polysaccharides, with variable structures and sizes (Pummer et al., 2013; Dreischmeier et al., 2017). Based on an estimation of mean molecular mass of soluble components, from mass concentration and osmolarity of the solution, Murray et al. (2022) suggested that the INMs comprise only a small proportion of the soluble material extracted from pollen. The soluble content of pollen solutions could be separated by size which would help to isolate the INMs but still larger volumes of pollen would be needed to produce concentrations sufficient for this analysis.

It is worth noting that the *Pinus mugo* collection results, where pollen from this species was collected in different locations in different years and demonstrates remarkably consistent IN

activity, does imply that using the same concentration of pollen in suspension can represent concentration of INMs, at least within a species.

To reflect on this more clearly in the text, as well as related points made by Prof. Grothe, we intend to add the following section to Section 2.10:

"Animal-pollinated plants can afford to allocate a lesser proportion of their resources to pollen production than wind-pollinated plants; plant-pollinator relationships ensure a higher likelihood that an individual pollen grain will reach the stigma of another plant and result in fertilisation (Herrera and Pellmyr, 2002). Therefore, it is challenging to collect large quantities of pollen from many animal-pollinated species without extensive harvesting of flowers. In this study, we prioritised nucleation temperature measurements which allowed a broad comparison of pollen from diverse plant species to be made. Limits on pollen quantities for some species restricted possibilities for further analysis, such as evaluating the quantity and composition of soluble material in each of the pollen solutions.

We have controlled for the mass of pollen added to a volume of water and time-in-suspension in an attempt to ensure consistency across the pollen samples tested, but this is not a direct measure of the concentration of INMs in each solution. Murray et al. (2022) compared the mass of dried soluble material from solutions of Carpinus betulus and Betula pendula pollen solutions, finding that for both types approximately 0.7% of the total mass of the samples was soluble material from pollen. However, even the quantity of soluble material is a proxy for the quantity of the INM component, which is estimated to be far smaller (Murray et al., 2022). That the activity of Pinus mugo pollen collected from plants in different locations, in different years, was so similar would seem to indicate the measurements carried out are in some capacity representative of the true pollen activity. Ultimately, these ambiguities highlight, in our view, the importance of work towards identifying the structure of the molecules responsible for the IN activity in pollen."

[revised manuscript text omitted]

---

## Author Comment (AC2)

**Responses to RC2**

We thank Professor Grothe for his thoughtful comments and suggestions for improving the manuscript. The specific comments made (blue text) are each addressed below (black text) with corresponding changes to the manuscript text included (red text).

**Major comments**

When Pummer 2012 recognized that soluble macromolecules can trigger heterogeneous ice nucleation, they assumed polysaccharides being responsible. The same conclusion was later also drawn by other authors, e.g. Dreischmeier 2017 and Gute 2020. The reason is that in the FTIR spectra the bands of polysaccharides are so intense that they overlay all other signals. However, Pummer 2013 already detected protein signals in their detailed study by Raman and FTIR spectroscopy. Only recently Burkart 2021 found evidence that proteins are present and are the responsible INMs. The fact that proteins and polysaccharides are present in the same solution might account for inherent mixtures of both or even for glycoproteins as the important INMs. In contrast to FTIR spectroscopy, fluorescence spectroscopy can clearly differentiate the proteins. Therefore, I strongly recommend to add fluorescence-excitation-emission-maps to figure 5, in order to have more meaningful results.

Burkart et al. (2021) revealed the strength of protein signal in birch pollen washing water decreased with decreasing solution concentration, correlating with a reduction in IN activity. At present, despite this observed correlation, the behaviour of pollen solutions in response to physical and chemical tests supports that the IN activity of these solutions is not, or at least not exclusively, proteinaceous in nature. The IN activity of pollen solutions has been shown to be stable in response to high temperatures (Dreischmeier et al., 2017; Daily et al., 2022) and exposure to protein denaturants and digesting enzymes, including guanidinium chloride (Pummer et al., 2012) and trypsin (Pummer et al., 2015), contrasting sharply with the response of known ice nucleating proteins. More recently, the possibility that proteins play some role in pollens' activity has been considered and we acknowledge that this is not resolved. However, we argue that the current evidence for the role of proteins does not justify carrying out additional measurements aimed at protein detection to this study, where the focus is not on exploring the structure of these ice nucleators.

The distortion of fluorescence signals in birch pollen solution, discussed by Seifried et al. (2022), also suggests that fluorescence spectroscopy cannot be used straightforwardly to compare protein content in pollen solutions which comprise complex mixtures of biomolecules. Seifried et al. (2022) attribute shifts in the wavelength of protein signals for a single pollen solution to concentration-dependant quenching or filter effects; indicating that the capacity for fluorescence measurements without additional analyses to compare between different pollen solutions, where the concentration of components is unknown and likely to vary, is severely limited.

We recognise that IR spectroscopy does not provide a comprehensive analysis of the content of pollen solutions. The structure of the INMs responsible for the IN activity of pollen remains ambiguous and undoubtedly warrants further investigation. The aims of this study were to test the IN activity of pollen from a broad range of species and to identify trends in the measured activity against various plant and pollen features; while we include some discussion of the INMs and have included FTIR spectra for three pollen solutions, detailed analyses of composition is ultimately out of the scope of this study, in particular because limits on available pollen quantities for some samples meant measurements had to be focussed in order to

achieve such a large comparison and identifying the active component of these solutions remains a difficult problem in its own right.

In figure 3 the authors have correlated the representative nucleation temperature with biological and geographical parameters and show that the impacts of these are not significant. However, when comparing fig 4 with the results in figure 3f, it becomes obvious that the growth elevation is only categorized in three classes, the selection of which is not clear. In literature, at least 4 categories are known, i.e. mountain zones 0-1800m, 1800-2300m, 2300-3000m and 3000-xm. When applying these mountain zone categories then a difference might become visible showing the change from alpine to snow zone being related to a significant increase of the nucleation temperature. In general, I appreciate such correlation boxplots. However, I wonder that the authors did only correlate representative nucleation temperature but did not also correlate other important parameters such as extractable amount of INMs, average mass of the INMs, size of the INMs, sizes of the aggregates of the INMs or even the intensity of the fluorescence signal (related to the protein concentration). This would significantly enhance the information value of the paper.

For Figure 3, species were divided into three plant growth elevation categories, described in Section 2.7 (L 219-223). It is important to highlight that the data available for plant growth elevations was limited, particularly for some rare species tested in this study. Nonetheless, limits for upper and lower possible growth elevations were assigned as shown in Table S1 based on the information available.

In contrast to the boxplot categorisation based primarily on the upper limit of plant growth elevation, Figure 4 shows the mean representative nucleation temperature for the full range of possible growth elevations, which have been binned into groups for every 50 m. For example, for the 1000 m bin, the representative nucleation temperature of all species which have a minimum elevation less than 1000 m and a maximum elevation greater than 1000 m are used in the calculation of the mean (and standard error). It must therefore be noted that some species fall into multiple bins. This figure shows mean representative nucleation temperature for elevations up to 3000 m is consistent. After 3000 m there seems to be a positive relationship however due to the small number of species in this range this is likely biased by an outlier.

As far as we are aware there is no standard categorisation for growth elevation based on upper and lower elevation limits – we could not find by searching the reference for the four mountain zones as defined by the reviewer. Using the values for upper elevation listed in Table S1 to categorise species into the four ranges suggested, gives the following figure:

[Figure]

(f) Categorised growth elevation

Critically, echoing the analysis of Figure 4 in L 233-237, the number of data points in the highest elevation category (upper growth elevation limit >3000 m) is too small to include in a statistical test. The ANOVA test failed to reject that there is a significant difference in mean representative nucleation temperature between 0-1800 m, 1800-2300 m, 2300-3000 m elevation categories (F = 0.655, p = 0.525), therefore we cannot determine a relationship between IN activity and these categories. This is not to say that there isn't a relationship between IN activity and plant growth elevation in nature but rather that from our data, even using these new categories, a relationship cannot be determined.

We agree that it would be valuable to consider the relationships between the selected plant features and additional parameters, such as the quantity and size distribution of soluble components. Unfortunately, we were limited by sample volumes. We prioritised nucleation temperature measurements for this study which allowed for a broad comparison of pollen from diverse plant species.

**Minor comments**

The term "INMs" has been coined by Pummer 2012 as "ice nucleating macromolecules". Please add "macro" when referring to this definition.

We appreciate the importance of keeping definitions clear and consistent in the literature and will change this in the text as advised.

INM mg$^{-1}$ extracted pollen was the general value. Unfortunately, this is not a very precise value since pollen have different amounts of extractable material on their surface (see Burkart 2021). More precise would be to determine the amount of soluble material in the solution subsequently to the extraction process by evaporating the water (or at least to show that the difference between mg$^{-1}$ Pollen and mg$^{-1}$ solute is neglectable).

We agree that determining the mass of soluble material in the pollen solutions would be of interest, particularly to compare between structurally diverse pollen. Unfortunately, due to limits on pollen volume it was not possible to measure this for the range of species in this study.

To reflect on this more clearly in the text, as well as related points made by Prof. Morris, we intend to add the following section to Section 2.10:

"Animal-pollinated plants can afford to allocate a lesser proportion of their resources to pollen production than wind-pollinated plants; plant-pollinator relationships ensure a higher likelihood that an individual pollen grain will reach the stigma of another plant and result in fertilisation (Herrera and Pellmyr, 2002). Therefore, it is challenging to collect large quantities of pollen from many animal-pollinated species without extensive harvesting of flowers. In this study, we prioritised nucleation temperature measurements which allowed a broad comparison of pollen from diverse plant species to be made. Limits on pollen quantities for some species restricted possibilities for further analysis, such as evaluating the quantity and composition of soluble material in each of the pollen solutions.

We have controlled for the mass of pollen added to a volume of water and time-in-suspension in an attempt to ensure consistency across the pollen samples tested, but this is not a direct measure of the concentration of INMs in each solution. Murray et al. (2022) compared the mass of dried soluble material from solutions of *Carpinus betulus* and *Betula pendula* pollen solutions, finding that for both types approximately 0.7% of the total mass of the samples was soluble material from pollen. However, even the quantity of soluble material is a proxy for the quantity of the INM component, which is estimated to be far smaller (Murray et al., 2022). That the activity of *Pinus mugo* pollen collected from plants in different locations, in different years, was so similar would seem to indicate the measurements carried out are in some capacity representative of the true pollen activity. Ultimately, these ambiguities highlight, in our view, the importance of work towards identifying the structure of the molecules responsible for the IN activity in pollen."

INMs have not only be found on pollen but also on leaf, bark, stem and branches (see Felgitsch 2018).

We acknowledge this point and intend to add this reference to the Introduction (L 29-33):

"The identification of ice nucleators associated with various organisms, from bacteria (Lindow et al., 1982; Maki et al., 1974; Lukas et al., 2022) to lichens (Kieft, 1988; Eufemio et al., 2023) and plants (Gross et al., 1988; Brush et al., 1994; Felgitsch et al., 2018), is of biological and atmospheric interest."

As well as adding the following sentence to L 42:

"Other plant parts including wood and leaves have also been found to host INMs (Felgitsch et al., 2018)."

**References**

Brush, R. A., Griffith, M., and Mlynarz, A.: Characterization and Quantification of Intrinsic Ice Nucleators in Winter Rye (Secale cereale) Leaves', Plant Physiol, 104, 725–735, 1994.

Burkart, J., Gratzl, J., Seifried, T., Bieber, P., and Grothe, H.: Subpollen particles (SPP) of birch as carriers of ice nucleating macromolecules, Biogeosciences Discuss., 1–15, 2021.

Daily, M. I., Tarn, M. D., Whale, T. F., and Murray, B. J.: An evaluation of the heat test for the ice-nucleating ability of minerals and biological material, Atmos. Meas. Tech., 15, 2635–2665, https://doi.org/10.5194/amt-15-2635-2022, 2022.

Dreischmeier, K., Budke, C., Wiehemeier, L., Kottke, T., and Koop, T.: Boreal pollen contain ice-nucleating as well as ice-binding "antifreeze" polysaccharides, Sci. Rep., 7, 1–13, https://doi.org/10.1038/srep41890, 2017.

Eufemio, R. J., de Almeida Ribeiro, I., Sformo, T. L., Laursen, G. A., Molinero, V., Fröhlich-Nowoisky, J., Bonn, M., and Meister, K.: Lichen species across Alaska produce highly active and stable ice nucleators, Biogeosciences, 20, 2805–2812, https://doi.org/10.5194/BG-20-2805-2023, 2023.

Felgitsch, L., Baloh, P., Burkart, J., Mayr, M., Momken, M. E., Seifried, T. M., Winkler, P., Schmale, D. G., and Grothe, H.: Birch leaves and branches as a source of ice-nucleating macromolecules, Atmos. Chem. Phys., 18, 16063–16079, https://doi.org/10.5194/acp-18-16063-2018, 2018.

Gross, D. C., Proebsting, E. L., and Maccrindle-Zimmerman, H.: Development, Distribution, and Characteristics of Intrinsic, Nonbacterial Ice Nuclei in Prunus Wood, Plant Physiol., 88, 915–922, https://doi.org/10.1104/PP.88.3.915, 1988.

Herrera, C. M. and Pellmyr, O.: Plant Animal Interactions: An Evolutionary Approach, John Wiley & Sons, 2002.

Kieft, T. L.: Ice nucleation activity in lichens, Appl. Environ. Microbiol., 54, 1678–1681, https://doi.org/10.1128/AEM.54.7.1678-1681.1988, 1988.

Lindow, S. E., Arny, D. C., and Upper, C. D.: Bacterial Ice Nucleation: A Factor in Frost Injury to Plants, Plant Physiol., 70, 1084–1089, https://doi.org/10.1104/PP.70.4.1084, 1982.

Lukas, M., Schwidetzky, R., Eufemio, R. J., Bonn, M., and Meister, K.: Toward Understanding Bacterial Ice Nucleation, J. Phys. Chem. B, 126, 1861–1867, https://doi.org/10.1021/ACS.JPCB.1C09342/ASSET/IMAGES/LARGE/JP1C09342_0004.JPEG, 2022.

Maki, L. R., Galyan, E. L., Chang-Chien, M.-M., and Caldwell, D. R.: Ice nucleation induced by pseudomonas syringae, Appl. Microbiol., 28, 456–459, https://doi.org/10.1128/AM.28.3.456-459.1974, 1974.

Murray, K. A., Kinney, N. L. H., Griffiths, C. A., Hasan, M., Gibson, M. I., and Whale, T. F.: Pollen derived macromolecules serve as a new class of ice-nucleating cryoprotectants, Sci. Reports 2022 121, 12, 1–11, https://doi.org/10.1038/s41598-022-15545-4, 2022.

Pummer, B. G., Bauer, H., Bernardi, J., Bleicher, S., and Grothe, H.: Suspendable macromolecules are responsible for ice nucleation activity of birch and conifer pollen, Atmos. Chem. Phys., 12, 2541–2550, https://doi.org/10.5194/acp-12-2541-2012, 2012.

Pummer, B. G., Budke, C., Augustin-Bauditz, S., Niedermeier, D., Felgitsch, L., Kampf, C. J., Huber, R. G., Liedl, K. R., Loerting, T., Moschen, T., Schauperl, M., Tollinger, M., Morris, C. E., Wex, H., Grothe, H., Pöschl, U., Koop, T., and Fröhlich-Nowoisky, J.: Ice nucleation by water-soluble macromolecules, Atmos. Chem. Phys., 15, 4077–4091, https://doi.org/10.5194/acp-15-4077-2015, 2015.

Seifried, T. M., Bieber, P., Weiss, V. U., Pittenauer, E., Allmaier, G., Marchetti-Deschmann, M., and Grothe, H.: Fluorescence signal of proteins in birch pollen distorted within its native

matrix: Identification of the fluorescence suppressor quercetin-3-O-sophoroside, Anal. Bioanal. Chem., https://doi.org/10.1007/s00216-022-04109-0, 2022.

---

## Author Response (AR2)

Dear Dr Stoy,

We would like to acknowledge the suggestions from Professor Hinrich Grothe in response to our revised manuscript entitled *'High Interspecific Variability in Ice Nucleation Activity Indicates Pollen Ice Nucleators are Incidental'* submitted to *Biogeosciences.* We appreciate the important distinctions highlighted by Prof. Grothe and his thorough review of our study. In response to these points, we have removed the specific use of "polysaccharides" when referring to the ice nucleating macromolecules, added statements acknowledging that proteins are present in pollen solutions and highlighted that their further study would help to shed light on the role they play in the ice nucleation activity. The specific changes made to the manuscript are outlined below.

**Line 20** "The results suggest that a polysaccharide present in pollen is produced by plants for a purpose unrelated to ice nucleation and has an incidental ability to nucleate ice." is replaced with "The results suggest that these macromolecules are produced by plants for a purpose unrelated to ice nucleation and have an incidental ability to nucleate ice."

**Line 55** "Both the chemical nature and biological function of these INMs remains unclear, although it seems very likely that they are polysaccharides (Dreischmeier et al., 2017)" is replaced with: "Both the chemical nature and biological function of these INMs remain unclear. Growing evidence suggests that polysaccharides are not exclusively responsible for pollens' ability to nucleate ice. Burkart et al. (2021) and Wieland et al. (2024) present evidence that proteins in *Betula pendula* pollen solution play a role in its IN activity."

**Line 280** "polysaccharides" is replaced with "soluble material".

**Line 296** "It must also be noted that proteins are present in pollen solutions (Pummer et al., 2013; Burkart et al., 2021) and associated absorption signals may be masked by stronger polysaccharide absorbances. Further investigation of the structure and variation of proteinaceous material across species' pollen would be of value to disentangle the role of protein and polysaccharide constituents in the measured IN activity." is added.

**Line 446, Line 448** and **Line 457** "polysaccharides" is replaced by "INMs".

**Line 459** "…large polysaccharides or polysaccharide aggregates…" is replaced by "… large molecules or molecular aggregates…".

**Line 462** and **Line 464** "polysaccharide" is replaced by "macromolecule".

We thank the reviewers and the editorial team for facilitating an engaging discussion and improvements to our manuscript and we hope that these changes adequately address the suggestions made.

Yours sincerely,

Nina Kinney and Tom Whale